# 🦜 PARROT : Zero-Shot Narrative Reading Comprehension via Parallel Reading

**Chao Zhao**      **Anvesh Rao Vijjini**      **Snigdha Chaturvedi**

Department of Computer Science, UNC Chapel Hill

{zhaochao, anvesh, snigdha}@cs.unc.edu

## Abstract

Narrative comprehension is a challenging task that requires a deep understanding of the foundational elements of narratives. Acquiring this skill requires extensive annotated data. To mitigate the burden of data annotation, we present PARROT , a zero-shot approach for narrative reading comprehension through parallel reading, which involves two parallel narratives that tell the same story. By leveraging one narrative as a source of supervision signal to guide the understanding of the other, PARROT abstracts the textual content and develops genuine narrative understanding. Evaluation conducted on two narrative comprehension benchmarks demonstrates that PARROT surpasses previous zero-shot approaches and achieves comparable performance to fully supervised models. The code will be available at https://github.com/zhaochaocs/Parrot.

## 1 Introduction

Narratives have long been recognized as a valuable resource for linguistic, scientific, cultural, and social learning (Rosen, 1985; Knoespel, 1991; Lyle, 2000; Nash, 2005; Bettelheim, 2010). Narrative comprehension, therefore, is considered a fundamental aspect of human intelligence (Bruner, 1997) and an important tool for cognitive development and meaning-making (Polkinghorne, 1988). With this motivation, previous research has tackled the task of narrative reading comprehension, which involves automatically comprehending a given narrative and answering questions related to it (Hirschman et al., 1999; Richardson et al., 2013).

However, in comparison to general text comprehension, which typically focuses on the understanding of named entities and factual information (Rajpurkar et al., 2016), narrative comprehension presents unique challenges. Specifically, it requires understanding the foundational elements of narratives. These elements include events along with their temporal and causal connections; settings

Figure 1: Illustration of parallel reading. $\mathcal{N}$ and $\mathcal{N}^+$ are different renderings of the same story. The key idea is to ask questions from $\mathcal{N}$ and encourage the model to answer them from $\mathcal{N}^+$. This helps the model in learning deep comprehension skills (as indicated in []).

such as the time, place, and environment; as well as characters, including their motivations, desires, emotions, and relationships with other characters. Together they exhibit intricate plot structures and involve complex character interactions, making it challenging for machines to comprehend. Despite the availability of extensively annotated data for general text reading comprehension, there is currently a lack of sufficient annotated data in the narrative domain, and it is not optimal to directly use models trained on general text data for narrative reading comprehension. Hence, there is a need to develop data-efficient learning approaches for narrative reading comprehension.

To address the aforementioned challenges, our idea is to leverage *parallel reading*: reading two parallel narratives that convey the same story but differ in various aspects of story-telling style. This

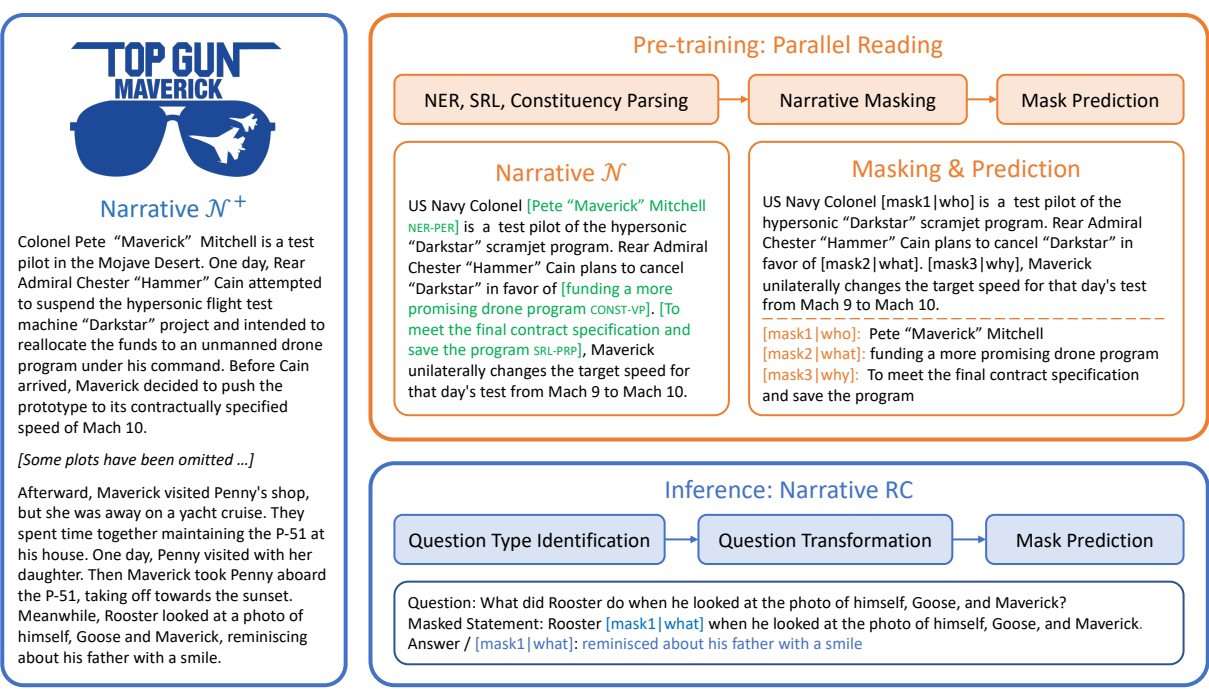

Figure 2: Illustration of the proposed approach, PARROT. During pre-training, we collect two parallel narratives, $\mathcal{N}^+$ and $\mathcal{N}$. We mask narrative-specific spans in $\mathcal{N}$ and pre-train the model to predict these spans by reading $\mathcal{N}^+$. During inference, we transform the question into a masked statement, following the pre-training format. Then we apply the pre-trained model to predict the answer based on the narrative and the masked statement. Note that for illustrative purposes, $\mathcal{N}^+$ is shared between pre-training and inference, but in real scenarios, there is no overlap.

idea aligns with the classical model of narrative theory (Genette, 1983), which emphasizes the perspectival nature of narratives – narratives encompass not only the sequence of events (the *story*), but also the ordering, granularity, point-of-view, and localization (the *discourse* and *narrating*). Ideally, comprehending either narrative would result in the same understanding of the story. Therefore, we can teach the model to develop reading comprehension skills by asking questions based on one narrative and encouraging the model to answer them by reading the parallel narrative. Figure 1 illustrates this concept. We will explain later how we operationalize this idea of asking and answering questions through masked language modeling.

Learning from parallel reading offers two advantages. Firstly, by exposing the model to narrative variations of the same story, we discourage its reliance on text-matching and enhance its ability to comprehend paraphrases, integrate information from long contexts, and perform multi-hop reasoning (as seen in Q2 and Q3 in Figure 1). Secondly, one narrative may contain information that is not explicitly stated in the other narrative, but can be implicitly inferred through a deeper understanding of the context. Training a model to deduce

such implicit information empowers it to surpass superficial understanding and grasp the implicit information and underlying meaning within the narrative (as seen in Q4 and Q5 in Figure 1). These advantages have been demonstrated in pedagogy to improve students' reading comprehension abilities (Schumaker et al., 1984; Grellet, 1981; Yano et al., 1994; Sipe, 2001).

With this idea in mind, we propose PARROT [1], a novel pre-training approach for zero-shot narrative comprehension. Figure 2 shows an overall illustration. It selectively masks important narrative elements within one narrative, and then pre-trains the model to predict these masked elements by reading the parallel narrative. To encourage PARROT to learn about a wide array of narrative elements, we mask a diverse set of elements covering characters, events, time, place, environments, and more. Lastly, to enable PARROT to perform narrative reading comprehension in a zero-shot manner, we narrow the disparity between the pre-training task of span prediction and the downstream task of reading comprehension by aligning their data formats.

---

[1]Stands for **pa**rallel **r**eading for ze**ro**-sho**t** narrative comprehension.

We conducted experiments on two narrative reading comprehension benchmarks, narrativeQA (Kočiský et al., 2018) and FairytaleQA (Xu et al., 2022), to evaluate PARROT. The results demonstrate that without any human annotation, PARROT achieves performance that is comparable to that of a fully supervised model. Furthermore, PARROT exhibits superior performance compared to supervised models when applied to out-of-domain datasets, demonstrating its effectiveness in transfer learning scenarios.

Our contributions are three-fold:

- We present PARROT, a novel pre-training approach for effective zero-shot narrative comprehension.

- We introduce a novel *parallel reading* strategy that involves utilizing different versions of narratives during pre-training to foster genuine narrative understanding.

- Our approach achieves competitive or better performance when compared to supervised models, showcasing its effectiveness in narrative comprehension tasks.

## 2 Related Works

Data-efficient reading comprehension has been addressed by automatically generating (passage, question, answer) triples for pre-training a model (Lewis et al., 2019, 2021). However, it requires annotated data to train a question-generation module. One possible solution is to replace it with a heuristic-based question-generation module (Heilman and Smith, 2010; Li et al., 2020; Lyu et al., 2021). Another option is to replace natural questions with cloze questions by masking the answer from the passage (Hermann et al., 2015; Hill et al., 2015). Researchers have proposed masking and predicting recurring spans within the passage as answers (Bian et al., 2021; Ram et al., 2021). In contrast to these approaches that focus on general extractive reading comprehension, PARROT specifically addresses the unique challenge of open-ended narrative comprehension and leverages parallel reading for abstractive comprehension and free-form question answering.

## 3 PARROT

In narrative reading comprehension, the input is a narrative $\mathcal{N}$ and a question $q$, while the output is a concise answer $a$. We develop PARROT, a zero-shot solution for this problem. PARROT utilizes a masked language modeling (MLM) based pre-training approach, which incorporates a selective span masking strategy to mask essential narrative elements (Sec. 3.1) and a parallel reading strategy to learn to predict the masked spans (Sec. 3.2). Next, to utilize the pre-trained model in a zero-shot fashion, we transform the downstream narrative reading comprehension task to match the format of the pre-training task (Sec. 3.3). Figure 2 shows an overall illustration.

## 3.1 Selective Span Masking

In model pre-training, a commonly used technique is masked language modeling (MLM), where spans are randomly masked for the model to predict (Devlin et al., 2019; Raffel et al., 2020). In previous works on pre-training for reading comprehension, named entities and recurring spans were masked as they are more closely associated with factual information (Ram et al., 2021; Bian et al., 2021). However, for narrative comprehension, the model needs to understand not just named entities but also various other narrative elements such as events, causality, temporal relationships, environmental settings, characters, their desires, personality traits, and relationships with others, to name a few. Previous masking strategies do not cover all these essential elements adequately.

Therefore, to enhance the model's ability to comprehend narratives, we incorporate a diverse set of masked spans to encourage the learning of a wide range of comprehension skills specific to narratives. We carefully select three types of spans to mask.

- Named entities: Named entities play a crucial role in narratives as they help identify characters and settings (such as time and place) within the narrative. We choose nine types of named entities: [2] Person, Location, Geopolitical Entity, Facility, Organization, Time, Date, Event, and Products.

- Semantic roles: Named entities alone can not encompass all narrative elements, such as settings like *last week* and *a small town*, event causality, characters' purpose, and more. Since these narrative elements usually unfold along with events, we focus on the associated arguments of verbs and include five semantic roles: [3] Direction (ARGM-DIR), Location

---

[2]We employ Spacy for NER. https://spacy.io/

[3]We employ AllenNLP for SRL. https://allenai.org/allennlp

(ARGM-LOC), Time (ARGM-TMP), Purpose (ARGM-PRP), Cause (ARGM-CAU), and Manner (ARGM-MNR).

- Verb and adjective phrases: A narrative can be seen as a sequence of events organized by a narrator (Schank and Abelson, 2013). To directly comprehend events, we identify verb phrases using constituency parsing [4] and mask them. Additionally, we mask adjective phrases to enhance the understanding of narrative settings and character characterization.

Given a narrative $\mathcal{N}$, we first identify and mask some spans, $m(\mathcal{N})$, and then pre-train a sequence-to-sequence model to predict these spans using the remaining text, $\mathcal{N}_{\backslash m(\mathcal{N})}$, and the original narrative, $\mathcal{N}$. We refer to this model as PARROT$_{\text{single}}$. The loss function is

$$\mathcal{L}_{\text{single}} = -\log p\left(m(\mathcal{N}) \mid \mathcal{N}_{\backslash m(\mathcal{N})}, \mathcal{N}\right). \quad (1)$$

## 3.2 Parallel Reading

Predicting masked spans using the original narrative $\mathcal{N}$ may result in the model trivially relying on the superficial lexical overlap. To mitigate this issue, we propose "parallel reading". Instead of solely masking and predicting spans within a single narrative $\mathcal{N}$, we leverage an additional parallel narrative, denoted as $\mathcal{N}^+$, which tells the same story as $\mathcal{N}$ but differs in granularity, point-of-view, etc. By simultaneously reading $\mathcal{N}$ and $\mathcal{N}^+$ and predicting the masked key information, we encourage the model to abstract the textual content and foster a genuine and deeper comprehension of narratives, avoiding overreliance on superficial textual matching clues. For example, in Figure 2, predicting [mask2] and [mask3] in $\mathcal{N}$ based on $\mathcal{N}^+$ requires more advanced comprehension skills, such as understanding character motivation and event causality.

Here we provide more details regarding parallel reading. Without loss of generality, we assume that $\mathcal{N}^+$ is longer than $\mathcal{N}$. We selectively mask spans in the shorter narrative, $\mathcal{N}$, and utilize the longer narrative, $\mathcal{N}^+$, as a source of evidence to predict the masked spans, since the longer narrative is likely to contain the necessary information present in the shorter narrative.

However, $\mathcal{N}$ might also contain some spans that are not answerable from $\mathcal{N}^+$. Masking such spans can result in noise in the training data. To mitigate

---

[4] We employ AllenNLP for constituency parsing.

this noise, we apply two filtering steps: one at the sentence level and another at the span level. At the sentence level, for each sentence $s$ in $\mathcal{N}$, we require the Rouge-1 Precision score (Lin, 2004) between $s$ and $\mathcal{N}^+$ to surpass a predefined threshold. If it does not, we do not mask spans from $s$. This criterion ensures that the remaining sentences in $\mathcal{N}$ align closely with the corresponding content in $\mathcal{N}^+$. At the span level, we selectively mask spans in $\mathcal{N}$ that directly or indirectly appear in $\mathcal{N}^+$. For spans that correspond to a named entity, we verify their presence in $\mathcal{N}^+$ using exact match. For spans that correspond to semantic roles or constituency phrases, which are more likely to be paraphrased, we adopt a more lenient criterion. For them, we calculate the Rouge-1 Precision score between the span and $\mathcal{N}^+$, setting a threshold to determine the acceptability of the span candidates for masking.

Lastly, we pre-train the model to predict the masked content within $\mathcal{N}$, given the concatenation of the masked narrative, $\mathcal{N}_{\backslash m(\mathcal{N})}$, together with the longer narrative, $\mathcal{N}^+$. The loss function is

$$\mathcal{L}_{\text{parallel}} = -\log p\left(m(\mathcal{N}) \mid \mathcal{N}_{\backslash m(\mathcal{N})}, \mathcal{N}^+\right). \quad (2)$$

## 3.3 Adapting to Reading Comprehension

In general, after the MLM pre-training, the pre-trained model requires fine-tuning with additional data to adapt to the specific downstream task. This fine-tuning is necessary because the pre-training task and the downstream task can be in different formats. However, in this paper, we do not assume access to the availability of any fine-tuning data and directly utilize the pre-trained model in a zero-shot manner. The key insight is that the reading comprehension task can be transformed into the MLM task. For example, in Figure 2, the question "*What did Rooster do when he looked at the photo?*" can be transformed into "*Rooster* [mask] *when he looked at the photo*", and the answer can be obtained by filling in the masked part. To achieve this transformation, we use QA2D (Demszky et al., 2018), which leverages a neural sequence model to generate masked statements from questions.

One drawback of this transformation strategy, as well as the pre-training strategy, is that the masked statement does not contain the question-type information typically conveyed by the wh-word in questions. For instance, without the original *what* question with the answer of "*reminisced about his father with a smile*", the masked statement in our example from Figure 2 can also be interpreted as

| Wh- Type | Span Type |
|---|---|
| Who | NER-PERSON |
| When | NER-TIME/DATE, ARGM-TMP |
| Where | NER-LOC/GEO/FAC, ARGM-LOC/DIR |
| Why | ARGM-CAU/PRP |
| How | ARGM-MNR, ADJP |
| What | Others |

Table 1: The mapping between the type of question and the corresponding type of masked span. This mapping enables the model to identify the appropriate type of question during pre-training.

a *how* question, leading to the possibility of filling the mask with a different answer such as "*felt delighted*".

To mitigate this ambiguity, we introduce a special type token preceding the mask to provide more accurate information about the question type. This token, as we illustrated in Figure 2, is typically a wh-element, such as *who* and *what*, which is extracted from the original question. To extract these words, we employ a constituency parser to parse the question and then identify the elements labeled with syntactic tags such as "WHNP", "WHADVP", "WHADJP", or "WHPP". During pre-training, since we lack the actual questions, we infer the question type based on the type of masked spans. The mapping between the span type and the question type is provided in Table 1.

By transforming the question to a masked statement during inference and incorporating the question type during pre-training, we establish a consistent data format for both pre-training and inference. This thereby empowers the model to perform zero-shot inference without explicit fine-tuning.

## 4 Experiments

In this section, we evaluate the performance of PARROT .

### 4.1 Datasest

**Datasets for Pre-training:** For parallel reading in the pre-training phase, we utilize NarraSum (Zhao et al., 2022), a dataset of 122K parallel narrative pairs obtained from plot descriptions of movies and TV episodes. After processing, we obtain a total of 57.4K paired narratives and 154.5K question-answer pairs. The average lengths of $\mathcal{N}$ and $\mathcal{N}^+$ are 125 and 926 tokens, respectively. Each narrative pair includes 2.7 masked spans on average.

To reduce input length and enhance computational efficiency, we partition the shorter narrative

$\mathcal{N}$ into smaller segments and predict the spans within each segment separately. However, we also need to strike a balance as excessively short segments would increase the overall number of training instances. Therefore, we opt to divide $\mathcal{N}$ into segments based on every three sentences.

**Datasets for Evaluation:** To evaluate the performance of PARROT , we conduct experiments on two narrative reading comprehension benchmarks: NarrativeQA (Kočiskỳ et al., 2018) and FairytaleQA (Xu et al., 2022). Since PARROT is zero-shot, we solely use the test sets of these datasets for evaluation. The narratives in FairytaleQA are derived from children's stories, while the narratives in NarrativeQA consist of plot summaries from books and movie scripts. For NarrativeQA, to avoid any potential overlap with the pre-taining data, we only consider instances derived from books for evaluation purposes. The average length of the narratives in these datasets is 150 and 659 tokens, respectively, and their test sets contain 1,007 and 10,557 question-answer pairs, respectively.

### 4.2 Setup

**Implementation Details**: The underlying model in PARROT is a T5-base (Raffel et al., 2020). We chose T5 because it has been pre-trained on a similar MLM task. Furthermore, compared with other MLM-based pre-trained models such as BART, T5 only predicts the masked tokens, making it more computationally efficient. During pre-training, we employ the AdamW optimizer (Loshchilov and Hutter, 2017) with a learning rate of $3 \times 10^{-5}$ and a batch size of $512$. We choose a large batch size because the pre-training data can be noisy. We incorporate warmup for the first 50 steps and implement early stopping based on the model's performance on the validation set. Training the models is conducted on four Tesla 3090 GPUs with 24 GB memory, taking approximately 4 hours to complete the pre-training process.

**Baselines:** Our first baseline is an information retrieval (IR) baseline adopted by Kočiskỳ et al. (2018), which selects the most similar sentence in the narrative to the given question and considers it as the answer. For computing this similarity, we use TF-IDF based cosine similarity. To establish stronger baselines, we compare PARROT with the model described in Lewis et al. (2021), which automatically generates question and answer pairs from the narrative. This involves utilizing an answer

|  | FairytaleQA | | | NarrativeQA | | |
|---|---|---|---|---|---|---|
|  | Rouge-1 | Rouge-2 | Rouge-L | Rouge-1 | Rouge-2 | Rouge-L |
| T5 Finetuned on FairytaleQA | 54.64 | 40.43 | 54.03 | 45.29 | 25.73 | 44.59 |
| T5 Finetuned on NarrativeQA | 49.64 | 38.45 | 49.10 | 65.05 | 36.04 | 64.49 |
| IR (TF-IDF) (Kočiskỳ et al., 2018) | 21.64 | 14.30 | 20.82 | 16.61 | 7.79 | 15.67 |
| AE-QG (Lewis et al., 2021) | 43.29 | 29.81 | 42.89 | 53.61 | 28.11 | 53.27 |
| Vicuna-13B (Chiang et al., 2023) | 37.52 | 20.44 | 35.98 | 32.59 | 17.37 | 31.46 |
| ChatGPT | 44.32 | 27.10 | 43.49 | 41.27 | 24.63 | 40.07 |
| PARROT$_{single}$ | 40.32 | 30.35 | 40.01 | 50.01 | 26.78 | 49.60 |
| PARROT | **48.56** | **36.83** | **48.10** | **55.71** | **30.81** | **55.32** |

Table 2: Results evaluated on FairytaleQA and NarrativeQA by Rouge scores. PARROT outperforms all baselines and achieves comparable or superior performance compared to supervised models in the out-of-domain setting.

extraction (AE) model and a question generation (QG) model trained on three MRC datasets: NQ (Kwiatkowski et al., 2019), TriviaQA (Joshi et al., 2017), and SQuAD (Rajpurkar et al., 2016). With these models, we generate question and answer pairs from the narratives in NarraSum, and then train a reading comprehension model based on T5-base. We refer to this baseline as AE-QG.

Additionally, we compare PARROT with Chat-GPT [5], a state-of-the-art large language model, and Vicuna-13B (Chiang et al., 2023), one of its best open-source alternatives. To use these models, we use the instruction "Please generate a brief answer rather than a complete sentence to the following question based on the provided passage as evidence.", alongside the passage and question that are appended. [6]

Lastly, we compare with fine-tuned models. We fine-tune the T5-base model on the training sets of narrativeQA and FairytaleQA, resulting in two fine-tuned models. We treat these results as upper bounds due to their supervised nature.

**Evaluation Measure:** Following the official evaluation of the two benchmarks, we use Rouge scores (Lin, 2004) between the predicted and the gold answers to evaluate the models.

### 4.3 Results

Table 2 presents the performance of PAR-ROT and baselines on FairytaleQA and Narra-tiveQA datasets. PARROT exhibits superior performance, significantly surpassing all zero-shot baselines (approximate randomization (Noreen, 1989; Chinchor, 1992), $p < 0.01$) . Additionally, it achieves performance that is 89.0% and 85.8%

comparable to those of fully supervised upper-bounds in terms of Rouge-L (48.10 vs. 54.03 and 55.32 vs. 64.49). These results demonstrate the effectiveness of PARROT in narrative reading comprehension. When comparing the strategies of single and parallel reading, PARROT achieves significantly higher performance compared to its single-reading counterpart, PARROT$_{single}$ , on both datasets. This result emphasizes the crucial role of parallel reading in enhancing model performance.

We also compare PARROT to supervised models under the out-of-domain setting, i.e., training the supervised model on one dataset and evaluating it on another. These results are displayed in gray font in the table. PARROT demonstrates competitive performance on FairytaleQA (Rouge-L of 48.10 vs. 49.10) and superior performance on NarrativeQA (Rouge-L of 55.32 vs. 44.59). This further demonstrates that PARROT can acquire general narrative comprehension skills and effectively apply them to diverse narratives.

Among the large language model baselines, ChatGPT exhibits stronger performance than Vicuna-13B. AE-QG models also achieve strong performance. However, these models require additional training data for training the answer extraction and question generation components. Furthermore, the generated question-answer pairs may contain errors, which could potentially impact the model's overall performance during pre-training.

### 4.4 Human Evaluation

To obtain a more reliable assessment of the model performance, we further conduct a human evaluation via Amazon Mechanical Turk (AMT). We randomly select 100 test instances from the test sets of both datasets. For each instance, we show three independent annotators the question, correct

[5] https://chat.openai.com/
[6] We tried different instructions and select the best-performing one.

|            | FairytaleQA | NarrativeQA |
|------------|-------------|-------------|
| IR (TF-IDF) | 2.12 | 2.37 |
| AE-QG | 2.56 | 2.91 |
| Vicuna-13B | 2.36 | 2.61 |
| ChatGPT | 2.49 | 2.86 |
| PARROT$_{single}$ | 2.30 | 2.78 |
| PARROT | 2.71 | 3.10 |

Table 3: Results of human evaluation on FairytaleQA and NarrativeQA.

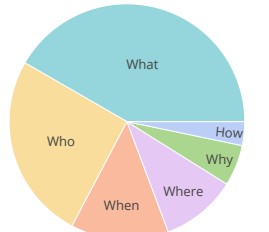 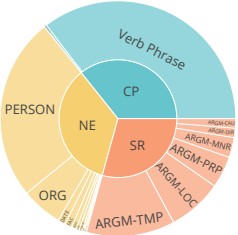

Figure 3: Distribution of the types of wh-elements and the sources of masked spans in pre-training data.

|        | FairytaleQA | NarrativeQA |
|--------|-------------|-------------|
| Random | 8.38 / 3.29 / 8.31 | 13.54 / 4.02 / 13.52 |
| None | 11.99 / 6.14 / 11.84 | 10.67 / 4.38 / 10.54 |
| + NE | 35.10 / 23.89 / 34.98 | 48.72 / 24.93 / 48.53 |
| + SR | 45.55 / 34.20 / 45.21 | 54.83 / 29.95 / 54.46 |
| + CP | 48.56 / 36.83 / 48.10 | 55.71 / 30.81 / 55.32 |

Table 4: The contribution of each source of masked spans to the final performance (R-1/R-2/R-L). We start with T5-base with and without further pre-training (Random and None). We then incrementally introduce named entities (NE), semantic roles (SR), and constituency phrases (CP) into the pre-trained data.

answers, and the answers generated by various systems. We then ask annotators to rate the quality of the predicted answers on a Likert scale ranging from 1 to 5. To maintain the evaluation quality, we require annotators to be AMT Masters based in the United States, with more than 1,000 HITs approved and an approval rate exceeding 98%. We manually review the annotation results, and if we identify annotators consistently providing low-quality annotations, we block them and re-assign their tasks. Annotators are compensated at a rate of $14 per hour, exceeding the local minimum wage.

Table 3 shows the results of human evaluation. The inter-annotator agreement score is 0.7003 in Gwet's gamma. Results from both datasets, along with the automatic measures, consistently demonstrate that Parrot outperforms the baseline models.

## 5 Analysis

We conduct analysis to better understand the behavior of PARROT .

### 5.1 Type of Masked Spans

One of our work's major contributions is incorporating a carefully selected and diverse set of masked spans geared toward narrative comprehension. To highlight the diversity, we analyze the distributions of different question types and the types of masked spans in the pre-training data. The results are presented in Figure 3. In terms of question types, the pre-trained data contains six major types: *what* (41.7%), *who* (25.6%), *when* (13.4%), *where* (10.4%), *why* (5.6%), and *how* (3.3%). In terms of masked spans, it shows that named entities (NE), semantic roles (SR), and constituency phrases (CP) are evenly distributed within the pre-training data. Specifically, named entities are predominantly represented by PERSON (72.0%) and ORG (16.1%) categories. Within semantic roles, Time (41.5%), Location (25.5%), and Purpose (14.9%) are the top three categories. Within the constituency phrases,

almost all of them fall under the category of verb phrases.

To investigate the impact of different mask types on the overall performance, we conduct an ablation study. During the construction of the pre-training data, we gradually expand the type of masked elements from named entities to semantic roles and constituency phrases. We also compare with a random masking strategy that aligns with the original pre-training objective of T5. The performance of the models trained on these versions of the pre-trained data is presented in Table 4. It reveals that focusing on named entities can improve the model performance, which aligns with previous research findings. However, relying solely on named entities is insufficient to encompass all narrative elements. By incorporating semantic roles, the model achieves a substantial improvement in performance. By including constituency phrases, we observe a further enhancement. On the contrary, when we continue pre-training with a random span masking strategy, we do not observe improvement in model performance. These results support our hypothesis that incorporating a diverse range of masked spans can significantly enhance models' ability of narrative comprehension.

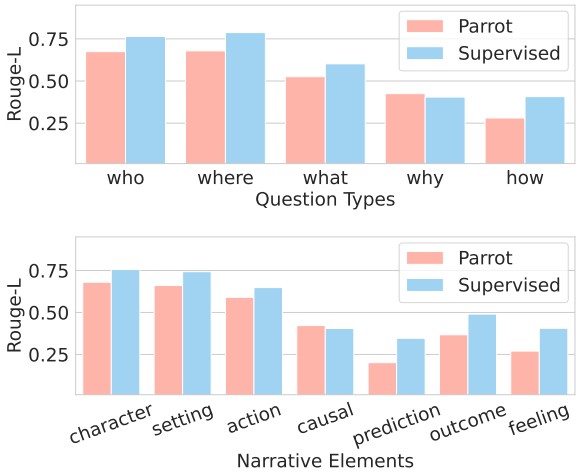

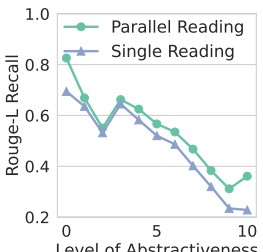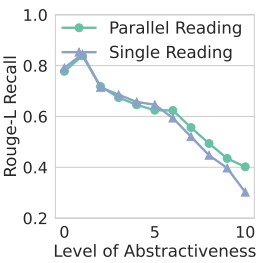

Figure 5: Model performance on FairytaleQA (left) and narrativeQA (right) w.r.t. the abstractiveness level between the question and the narrative. We report Rouge-L Recall to evaluate whether the correct answer is included in the predicted answer.

Figure 4: Fine-grained model performance on Fairy-tableQA w.r.t. the types of questions (top) and narrative elements (bottom).

## 5.2 Decomposition of Model Performance

We proceed to conduct a thorough analysis of the model's performance at a finer granularity. To accomplish this, we partition the FairytaleQA dataset into smaller subsets based on question types and narrative elements, as annotated within the dataset. Then we evaluate the model's performance on the individual subsets and compare it with the performance of the supervised model. The results are illustrated in Figure 4.

Comparing the results with the supervised model, PARROT demonstrates competitive performance in questions that involve identifying characters (*who*) and their activities (*what*), establishing causal relationships between events (*why*), and understanding the setting of the narrative (*where*). However, when dealing with more intricate narrative aspects such as pinpointing outcomes, predicting unknown events, and deciphering characters' emotional states (*how*), PARROT exhibits a larger performance gap. This particular strength and weakness align with the distribution of the types of masked spans present in the pre-training data. We leave enhancing the comprehension of these narrative components for future work.

## 5.3 Impact of Parallel Reading

In addition to incorporating a diverse array of narrative elements, a significant contribution of PARROT is leveraging parallel reading to abstract the textual content and comprehend the underlying meaning of the narrative. As discussed in Section 4.3, PARROT achieves better overall performance compared to PARROT$_{single}$ . In this section, we an-

alyze how parallel reading impacts the model's performance when the question is less lexically overlapped with the narrative.

To accomplish this, we divide the test set into subsets based on the level of abstractiveness between the question and the narrative. More specifically, we first identify the most similar sentence in the narrative with the question as the evidence sentence, and then use the sum of Rouge-1 precision and Rouge-2 precision between the question and the evidence sentence to approximate the level of abstractiveness. Higher Rouge Precision indicates lower abstractiveness. Figure 5 shows the model's performance based on the degree of abstractiveness between the question and the narrative.

The results demonstrate that, in general, as the question becomes increasingly abstractive (the right side of the $x$-axis), the performance gap between PARROT$_{single}$ and PARROT becomes more significant. This finding indicates that compared with PARROT$_{single}$ , PARROT is better at understanding abstractive questions and finding answers based on genuine comprehension, rather than mere text matching. It supports our motivation that parallel reading enhances the model's ability to comprehend the underlying meaning of the narrative.

Interestingly, in highly extractive scenarios (the left side of the $x$-axis), PARROT also outperforms its single-reading counterpart on FairytaleQA. This is because PARROT$_{single}$ tends to directly copy text from the narrative, which sometimes results in errors related to answer resolution when the copied part includes a pronoun instead of the proper entity mention. In contrast, PARROT is capable to select the appropriate entity mention as the answer, rather than mechanically copying the pronoun.

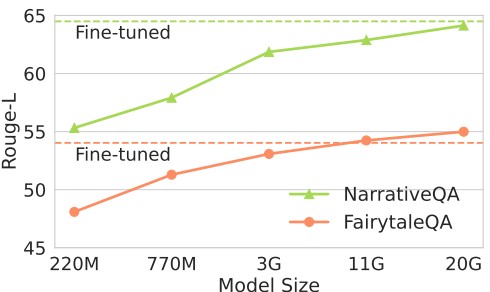

Figure 6: Model performance w.r.t. the size of underlying models.

## 5.4 Scaling to Larger Models

We further investigate the potential benefits of PARROT 's pre-training strategy for larger models. To this end, we experiment with different sizes of the underlying T5 model, namely base (220M), large (770M), XL (3B), XXL (11B), and UL2 20B (Tay et al., 2023), a T5-like large model. We fine-tune the entire parameters for T5 base and T5 large. For larger models, we utilize LORA (Hu et al., 2022) to facilitate more efficient fine-tuning due to the large computation. The results of these models are presented in Figure 6.

The results show that as we increase the size of the underlying model, the performance improves gradually, approaching and eventually surpassing the performance of the fine-tuned baselines. This experiment indicates that PARROT can effectively enhance a model's ability of narrative comprehension irrespective of its size.

## 5.5 Qualitative Analysis

Table 5 shows two examples of generated answers from different systems. To address the first question correctly, the model must distinguish that "Tsui" refers to the young man rather than "Rose-Red," who is his wife. Vicuna falls short in making this distinction. ChatGPT offers a consecutive answer that lacks useful information. In contrast, PARROT provides the correct answer. The second narrative is notably longer than the first, with the answer to the question not explicitly provided in the text. To answer this question, the model needs to establish a connection between the earlier plot when Ian requested the finest horse and the later plot where he won the race. AE-QG fails to establish this connection, resulting in a lack of useful information. Vicuna and ChatGPT partially answer the question by mentioning the race, but do not emphasize Ian's motivation to "win the race," which

---

**Narrative I:** Two years passed, and the youth no longer thought of any danger. Hence, when the flowers began to bloom in the spring, Rose-Red went driving in a small wagon outside the city, near the river. And there one of the prince's servants saw her, and informed his master. The latter sent for the youth, who, since he could not conceal the matter, told him the whole story exactly as it had happened. Said the prince: "The whole blame rests on Rose-Red. I do not reproach you. Yet since she is now your wife I will let the whole matter rest. But Molo will have to suffer for it!"

**Question:** Why did Tsui tell the prince the whole story exactly as it had happened?

**Model Outputs:**
**Vicuna-13B:** Rose-Red told the prince the whole story because the prince asked her to.
**ChatGPT:** To inform the prince of what had happened.
**AE-QG:** He could not conceal the matter.
**PARROT :** Because he could not conceal the matter.

**Reference:** He could not conceal the matter.

---

**Narrative II:** "That is a pity," returned the young man politely, but in his heart he rejoiced, for he wished to be alone. Leaving the house, he entered a grove of trees which stood behind, and took the box from his pocket. He raised the lid, and out flew the three little birds. "Good master, what shall we do for thee?" asked they, and he answered, "Bring me the finest horse that ever was seen, and the grandest dress, and glass shoes." "They are here, master," said the birds, and so they were, and never had the young man seen anything so splendid. Mounting the horse he rode into the ground where the horses were assembling for the great race, and took his place among them. Many good beasts were there which had won many races, but the horse of the fisher's son left them all behind, and he was first at the winning post. The king's daughter waited for him in vain to claim his prize, for he went back to the wood, and got off his horse, and put on his old clothes, and bade the box place some gold in his pockets. After that he went back to the weaver's house, and told him that the gold had been given him by the man who had won the race, and that the weaver might have it for his kindness to him.

**Question:** Why did Ian need the finest horse?

**Model Outputs:**
**Vicuna-13B:** Ian needed the finest horse for the race.
**ChatGPT:** To participate in the great race.
**AE-QG:** A pity.
**PARROT :** To win the race.

**Reference:** He wanted to win the horse-race.

---

Table 5: Sampled answers generated by different systems. For each example, we show the narrative, model predicted answers, and the referenced answer.

---

is the primary reason he sought the "finest" horse. Thanks to the long-term reasoning skills acquired during parallel pre-training, PARROT accurately answered this question.

## 6 Conclusion

We introduce PARROT , a novel zero-shot approach for narrative reading comprehension based on pre-training. By selectively masking significant elements within the narrative and pre-training the model to predict these spans through parallel reading, PARROT learns to abstract essential textual content and gains a genuine understanding of the narrative. Experimental results on two diverse narrative datasets demonstrate the superiority of PARROT , showcasing its effectiveness in enhancing narrative reading comprehension. Our analysis further emphasizes the significance of employing a diverse range of masked spans and leveraging the parallel reading strategy during model pre-training.

## Acknowledgements

This work was supported in part by NSF grant DRL-2112635. We thank anonymous reviewers for their thoughtful and constructive reviews.

## Limitations

One limitation of our approach, as discussed in the analysis section, is that the selective masking strategy employed in PARROT may not adequately encompass some intricate narrative elements such as event prediction and user emotion. Exploring these aspects is left as future work. Another limitation is that we rely on specific NLP modules for identifying the spans to be masked and transforming natural questions into masked statements. It is worthwhile to investigate the potential of employing large language models to replace these specialized modules.

Additionally, our focus in this paper is specifically on narrative question answering. While we have demonstrated the effectiveness of PARROT in this particular task, we have not extensively explored its performance in other narrative comprehension tasks such as summarization, or scenarios where questions are not well-formed. Finally, we only conduct experiments on English datasets. It would be beneficial to further explore our approach to other languages.

## Ethical Considerations

To address energy consumption concerns, we conduct the majority of experiments of PARROT using the relatively smaller T5-base model. We develop and test PARROT on publicly released datasets. Since we only pre-trained and tested on a few datasets, the developed model may exhibit incorrect answers for specific questions or demonstrate a bias towards certain types of narratives. We would encourage downstream users of PARROT to proactively anticipate and mitigate these potential risks.

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
