# OpenReview forum: "PARROT: Zero-Shot Narrative Reading Comprehension via Parallel Reading"
_EMNLP/2023/Conference — EMNLP 2023 Findings_

### Official Review · Reviewer_z7i5 · 2023-08-04

**Soundness:** 3

**Excitement:**

3: Ambivalent: It has merits (e.g., it reports state-of-the-art results, the idea is nice), but there are key weaknesses (e.g., it describes incremental work), and it can significantly benefit from another round of revision. However, I won't object to accepting it if my co-reviewers champion it.

**Missing References:**

This looks related to multi-view/consistency regularization work such as:
Semi-Supervised Sequence Modeling with Cross-View Training
Multi-View Sequence-to-Sequence Models with Conversational Structure for Abstractive Dialogue Summarization
Unsupervised Data Augmentation for Consistency Training

**Paper Topic And Main Contributions:**

This paper propose a pre-training framework, PARROT, for zero-shot narrative comprehension. Specifically, PARROT utilizes two versions of the same narrative and introduce a parallel reading strategy to predict masked spans (which are selected based on named entities, semantic roles, and verbs and adjective phrases) in one version based on another version during the pre-training to enhance the narrative understanding. Through experiments, their zero-shot performance match the supervised models.

**Questions For The Authors:**

1. Does the baseline also utilize the question transformation during the inference?
2. For PARROT-single pre-training, does it still utilize the all the narratives (i.e., 57.4*2K data)?


**Reasons To Accept:**

1. The parallel reading strategy which utilizes different renderings to help the understanding each other makes sense.
2. The zero-shot performance of PARROT outperforms or close to the supervised models and better than larger models.

**Reasons To Reject:**

1. Such strategy requires extra parallel data, which might not exist in many datasets/tasks especially during the pre-training stage. The authors did not consider such cases to propose some cheap ways to acquire such parallel data. Also, utilizing the parallel data for training increase the size of context window, which require larger context window for models and might be expensive.
2. The question answering requires the template mapping to transform the question into a masked statement, which might cause the poor generalization to questions that are not 'Wh-types'/transformable.


**Reproducibility:**

3: Could reproduce the results with some difficulty. The settings of parameters are underspecified or subjectively determined; the training/evaluation data are not widely available.

**Reviewer Confidence:**

5: Positive that my evaluation is correct. I read the paper very carefully and I am very familiar with related work.

---

> ### Author Rebuttal · Authors · 2023-08-29
>
> Thank you for your valuable feedback. We have addressed the comments outlined below and will incorporate these revisions in the next version.
>
> ---
>
> **Q1:** Such strategy requires extra parallel data, which might not exist in many datasets/tasks especially during the pre-training stage. The authors did not consider such cases to propose some cheap ways to acquire such parallel data.
>
> **A1:** Prior work has presented inexpensive ways to collect the kind of parallel data PARROT needs. For instance, in our research, we use the NarraSum dataset, which is a collection of extensive parallel data acquired in an automatic and inexpensive manner. Other studies have shown that collecting different types of parallel data on a larger scale is feasible, such as dialogue-narration pairs [1], query-document pairs [2], sentence simplification pairs [3], bilingual pairs [4], image-caption pairs [5], and more.
>
> [1] Storytelling with Dialogue: A Critical Role Dungeons and Dragons Dataset, ACL 2020
>
> [2] C-MORE: Pretraining to Answer Open-Domain Questions by Consulting Millions of References, ACL 2022
>
> [3] MUSS: Multilingual Unsupervised Sentence Simplification by Mining Paraphrases, LERC 2022
>
> [4] Unsupervised Bitext Mining and Translation via Self-Trained Contextual Embeddings, TACL 2020
>
> [5] Learning Transferable Visual Models From Natural Language Supervision, ICML 2021
>
> ---
>
> **Q2:** Also, utilizing the parallel data for training increase the size of context window, which require larger context window for models and might be expensive.
>
> **A2:** Increasing context is inevitable in reading comprehension. The model must have some extra content to encode the question besides the passage. In Parrot, the only difference is that the model encodes the masked parallel narrative instead of the question. Also, parallel training will not increase the size of the context window too much. As mentioned in lines 339 - 346, we only augment the context by three sentences ( a few dozen tokens), which should not pose a significant problem.
>
> ---
>
> **Q3:** The question answering requires the template mapping to transform the question into a masked statement, which might cause the poor generalization to questions that are not 'Wh-types'/transformable.
>
> **A3:** In this paper, we focus on “Wh” or “How” type questions. For other types of questions such as yes/no questions or multiple choice questions, the template mapping is straightforward. We can simply provide candidate options to mask and employ the Parrot model to calculate the probability of each candidate. In contrast, the 'Wh-question' is more challenging because the model must generate answers rather than selecting from a set of candidates.
>
> For questions that are not well-formed, the transformation process may introduce errors. Future work is needed to develop automatic transformation strategies to address these challenges, which is beyond the scope of this paper. We are happy to discuss them within the paper.
>
> ---
>
> **Q4:** Does the baseline also utilize the question transformation during the inference?
>
> **A4:** It depends on whether the baseline receives a question or a masked statement as input during pre-training. For the IR baseline, there isn't a substantial difference between these two inputs. AE-QG receives original questions as input due to the introduction of question generation during its pre-training. Vicunna and ChatGPT also utilize original questions as they are designed as conversational models. Parrot-single employs masked statements as input because its pre-training objective aligns with Parrot.
>
> ---
>
> **Q5:** For PARROT-single pre-training, does it still utilize the all the narratives (i.e., 57.4*2K data)?
>
> **A5:** To ensure a fair comparison, both Parrot and Parrot-single utilize 57.4k data for pre-training. We also explored pre-train the Parrot-single using all narratives, but the results were not significantly different. This is because the PARROT-single model only learns to copy from the original document to fill in masks through shallow text matching, which is a straightforward task for its underlying model, T5, to learn, therefore the additional training data does not yield substantial improvements.

---

### Official Review · Reviewer_bdc5 · 2023-08-05

**Soundness:** 3

**Excitement:**

3: Ambivalent: It has merits (e.g., it reports state-of-the-art results, the idea is nice), but there are key weaknesses (e.g., it describes incremental work), and it can significantly benefit from another round of revision. However, I won't object to accepting it if my co-reviewers champion it.

**Paper Topic And Main Contributions:**

The main contribution of the article is the proposal of PARROT, a pre-training method for narrative reading comprehension that leverages parallel narratives. This approach utilizes "Selective Span Masking" to cultivate the model's reading comprehension abilities in the narrative domain. This method aims to address the lack of data in the narrative domain, improve data efficiency, and enhance the model's capability for zero-shot narrative reading comprehension.

**Reasons To Accept:**

1.This paper introduces a novel pre-training task called PARROT, which utilizes parallel reading to enhance the model's capability for zero-shot narrative reading comprehension.

2.This paper's empirical experiments validate the significance of "Selective Span Masking".

**Reasons To Reject:**

1.The paper has a relatively limited comparison of the generated results. It can further enhance the comparison between the proposed method and baseline results using other methods such as Human Evaluation and Case Studies.

2.The authors should include a comparative model that continues pre-training using T5's original pre-training task on the author's pre-training dataset. Furthermore, a comprehensive analysis and comparison should be conducted between PARROT's parallel reading approach and the performance of this model.

**Reproducibility:**

4: Could mostly reproduce the results, but there may be some variation because of sample variance or minor variations in their interpretation of the protocol or method.

**Reviewer Confidence:**

4: Quite sure. I tried to check the important points carefully. It's unlikely, though conceivable, that I missed something that should affect my ratings.

---

> ### Author Rebuttal · Authors · 2023-08-29
>
> Thank you for your valuable feedback. We have addressed the comments outlined below and will incorporate these revisions in the next version.
>
> **Q1.** The paper has a relatively limited comparison of the generated results. It can further enhance the comparison between the proposed method and baseline results using other methods such as Human Evaluation and Case Studies.
>
> **A1.** For additional automatic evaluation metrics, we’ve incorporated both the Exact Match (EM) and F1 score. We also conducted a human evaluation using Amazon Mechanical Turk. We randomly selected 100 test instances from both test sets. For each instance, we show three independent turkers the question, correct answers, and the predicted answers. Turkers were asked to assess the quality of the predicted answers on a Likert scale ranging from 1 to 5. To maintain the evaluation quality, we require turkers to be AMT Masters based in the US, with more than 1,000 HITs approved and an approval rate exceeding 98%. We manually checked the annotation results, blocked turkers who consistently provided low-quality annotations, and re-distributed their tasks. Human judges received a wage rate of $14 per hour.  The IAA score is 0.7003 in Gwet’s gamma.
>
> |       |    EM     |    F1     |  Human  |
> |-------|:---------:|:---------:|:-------:|
> |   IR  | 0.20/0.02 | 19.69/15.12 | 2.12/2.37 |
> |  AE-QG | 22.74/34.90 | 41.62/52.93 | 2.56/2.91 |
> | Vicuna | 2.18/2.96 | 31.63/30.87 | 2.36/2.61 |
> | ChatGPT | 10.72/6.85 | 46.18/39.83 | 2.49/2.86 |
> | Parrot-single | 15.79/29.43 | 37.73/48.94 | 2.30/2.78 |
> |  Parrot  | 23.83/35.89 | 46.59/54.11 | 2.71/3.10 |
>
> Results on both datasets (separated by “/”) show that both extra automatic metrics and human evaluation support that Parrot is significantly better than baselines (approximate randomization p<0.05). We will add this and case studies to the paper.
>
> ---
>
> **Q2.** The authors should include a comparative model that continues pre-training using T5's original pre-training task on the author's pre-training dataset. Furthermore, a comprehensive analysis and comparison should be conducted between PARROT's parallel reading approach and the performance of this model.
>
> **A2.** T5 is already pre-trained on our dataset. The majority of our pre-training dataset is sourced from Wikipedia and online resources that are already included in C4, the pre-training data for T5. In Table 3, we provide a comparison between this T5 (indicated as “none”) and our "Selective Span Masking" strategy. The results demonstrate that our strategy significantly enhances model performance.
>
> We continue to pre-train T5 using the NarraSum dataset using T5's original pre-training task. We utilize two pre-training objectives. The first (M1, single-reading) is the same as T5's original pre-training objective, where we mask spans from a narrative and then train the model to predict these masked spans. For the second objective (M2, parallel reading), we mask spans from N and train the model to predict these spans based on both the masked N and the unmasked N+. Results on two datasets (separated by “/”) are presented below. It demonstrates that continuing pre-training without the selective span masking strategy cannot improve model performance.
>
> | Model | Rouge-1   | Rouge-2   | Rouge-L   |
> |-------|-----------|-----------|-----------|
> | M1    | 7.94/11.05 | 3.24/3.55 | 7.88/11.03 |
> | M2    | 8.38/13.54 | 3.29/4.02 | 8.31/13.52 |

---

### Official Review · Reviewer_jhYe · 2023-08-15

**Soundness:** 3

**Excitement:**

4: Strong: This paper deepens the understanding of some phenomenon or lowers the barriers to an existing research direction.

**Paper Topic And Main Contributions:**

This paper presents PARROT, a zero-shot approach for narrative reading comprehension through parallel reading. PARROT treats one narrative as supervision signal to guide the understanding of the other. PARROT also surpasses previously zero-shot approaches on two narrative comprehension benchmarks, and is comparable to fully supervised models based on ROUGE scores.

**Questions For The Authors:**

From line 47 - 49, you mentioned that there is still no sufficient data in the narrative domain, and it is not optimal to directly use models trained on general text data for narrative reading comprehension. However, from my understanding, PARROT also  uses a parallel narrative pairs for pre-training, is the data bottleneck the lack of annotated questions and answer pair, or the text?

**Reasons To Accept:**

- To adapt to the narrative comprehension setup, PARROT increases types of spans to mask that includes named entities, semantic roles, as well as verbs and adjective phrases.
- PARROT utilize parallel reading corpuses to do narrative reading in a zero-shot manner, which is quite interesting.

**Reasons To Reject:**

- For experiment results, almost all analysis are based on ROUGE scores. However, ROUGE score is known to have problems and can be easily gamed (Krishna et al. 2021; https://arxiv.org/abs/2103.06332). I am yet not convinced that PARROT is indeed better than the baselines. More automatic evaluation metric as well as human evaluation should be conducted.
- What does the question and answer look like? Figure 4 briefly mentioned the types but more examples should be added.

**Reproducibility:**

4: Could mostly reproduce the results, but there may be some variation because of sample variance or minor variations in their interpretation of the protocol or method.

**Reviewer Confidence:**

4: Quite sure. I tried to check the important points carefully. It's unlikely, though conceivable, that I missed something that should affect my ratings.

---

> ### Author Rebuttal · Authors · 2023-08-29
>
> Thank you for your valuable feedback. We have addressed the comments outlined below and will incorporate these revisions in the next version.
>
> ---
>
> **Q1:** For experiment results, almost all analysis are based on ROUGE scores. However, ROUGE score is known to have problems and can be easily gamed (Krishna et al. 2021; https://arxiv.org/abs/2103.06332). I am yet not convinced that PARROT is indeed better than the baselines. More automatic evaluation metric as well as human evaluation should be conducted.
>
> **A1:** For additional automatic evaluation metrics, we’ve incorporated both the Exact Match (EM) and F1 score. We also conducted a human evaluation using Amazon Mechanical Turk. We randomly selected 100 test instances from both test sets. For each instance, we show three independent turkers the question, correct answers, and the predicted answers. Turkers were asked to assess the quality of the predicted answers on a Likert scale ranging from 1 to 5. To maintain the evaluation quality, we require turkers to be AMT Masters based in the US, with more than 1,000 HITs approved and an approval rate exceeding 98%. We manually checked the annotation results, blocked turkers who consistently provided low-quality annotations, and re-distributed their tasks. Human judges received a wage rate of $14 per hour.  The IAA score is 0.7003 in Gwet’s gamma.
>
> |       |    EM     |    F1     |  Human  |
> |-------|:---------:|:---------:|:-------:|
> |   IR  | 0.20/0.02 | 19.69/15.12 | 2.12/2.37 |
> |  AE-QG | 22.74/34.90 | 41.62/52.93 | 2.56/2.91 |
> | Vicuna | 2.18/2.96 | 31.63/30.87 | 2.36/2.61 |
> | ChatGPT | 10.72/6.85 | 46.18/39.83 | 2.49/2.86 |
> | Parrot-single | 15.79/29.43 | 37.73/48.94 | 2.30/2.78 |
> |  Parrot  | 23.83/35.89 | 46.59/54.11 | 2.71/3.10 |
>
> Results on both datasets (separated by “/”) show that both extra automated metrics and human evaluation support that Parrot is significantly better than baselines (approximate randomization p<0.05).
>
> The ROUGE score is problematic when assessing lengthy outputs, as exemplified in the “long-form” QA paper pointed out by the review. LFQA concludes that ROUGE lacks reliability as they have observed that simply copying questions yields higher ROUGE scores than predicting answers, and predicting answers yields higher ROUGE scores than using reference answers.
>
> However, our study primarily focuses on the “short-form” QA problem, where ROUGE is a reliable measure. For instance, when the reference contains only a few tokens, a higher ROUGE score effectively demonstrates the system's superiority. Also, none of the misleading behaviors of ROUGE mentioned in the LFQA paper were observed in the two datasets we employed. For these reasons, ROUGE serves as the official evaluation metric in both datasets, and is the sole official measure employed in the FairytaleQA (2022) paper.
>
> ---
>
> **Q2:** What does the question and answer look like? Figure 4 briefly mentioned the types but more examples should be added.
>
> **A2:** We have some examples in Figures 1 and 2 in the paper. We will add more examples in the appendix.
>
> ---
>
> **Q3:** From line 47 - 49, you mentioned that there is still no sufficient data in the narrative domain, and it is not optimal to directly use models trained on general text data for narrative reading comprehension. However, from my understanding, PARROT also uses a parallel narrative pairs for pre-training, is the data bottleneck the lack of annotated questions and answer pair, or the text?
>
> **A3:** The primary bottleneck lies in the availability of annotated question-and-answer pairs. The raw narrative text should not be a limiting factor. We will clarify it in the next version.

---

### Meta-Review · Area_Chair_6fwY · 2023-09-23

**Recommendation:** 3

**Metareview:**

The paper proposes Parrot, a training method to improve narrative reading comprehension, where the reading comprehension task is focused on a story.

To train, Parrot finds two narratives of the same story, generates questions about the first, then trains the model to answer the questions using the second narrative as context.

The model is pretrained on parallel narratives from NarraSum, then evaluated "zero-shot" on d FairytaleQA and NarrativeQA. Results show improved results compared to Vicuna-13B and ChatGPT. Results are still worse than fine-tuning on the training set of each dataset (which is not surprising given the zero-shot evaluation), but get closer.

During the discussion period, the authors augmented their ROUGE-based evaluation with additional human evaluation, which was consistent with the automatic metric.

One thought for future work: the automatic question generation had a lot of moving pieces with engineered rules, makes, templates, question types, named-entity generators... etc. While this seems to work, I would expect all of this machinery to be replaceable with prompting for an LLM, making the method simpler and more robust.

---

### Decision · Program_Chairs · 2023-10-07

**Decision:**

Accept-Findings

**Comment:**

The paper proposes Parrot, a training method to improve narrative reading comprehension, where the reading comprehension task is focused on a story.

To train, Parrot finds two narratives of the same story, generates questions about the first, then trains the model to answer the questions using the second narrative as context.

The model is pretrained on parallel narratives from NarraSum, then evaluated "zero-shot" on d FairytaleQA and NarrativeQA. Results show improved results compared to Vicuna-13B and ChatGPT. Results are still worse than fine-tuning on the training set of each dataset (which is not surprising given the zero-shot evaluation), but get closer.

During the discussion period, the authors augmented their ROUGE-based evaluation with additional human evaluation, which was consistent with the automatic metric.

One thought for future work: the automatic question generation had a lot of moving pieces with engineered rules, makes, templates, question types, named-entity generators... etc. While this seems to work, I would expect all of this machinery to be replaceable with prompting for an LLM, making the method simpler and more robust.